# Proximal and distal middle cerebral artery diameter ratio and lenticulostriate artery infarction

Jun Sang Yoo[1], Jae Hyun Choi[2], Jae Young Park[1], Jeong Yun Song[1], Jun Young Chang[1], Dong-Wha Kang[1], Sun U. Kwon[1], Hang Jin Jo[2,3], Bum Joon Kim[1*]

1 Department of Neurology, Asan Medical Center, University of Ulsan, Seoul, South Korea, 2 Department of Mechanical engineering, POSTECH, Pohang, South Korea, 3 Division of Advanced Nuclear Engineering, POSTECH, Pohang, South Korea

* medicj80@hanmail.net

## Abstract

### Background

Lipohyalinotic degeneration (LD) and branch atheromatous disease (BAD) can contribute to subcortical infarctions in the lenticulostriate artery (LSA) territory. This study aimed to identify the association between the proximal and distal middle cerebral artery (MCA) diameter ratio and the two different pathomechanisms of LSA infarction.

### Methods

Patients with acute LSA infarctions categorized as small vessel occlusive disease were included. Demographic and clinical data, along with MCA geometrical variables, were collected. LD and BAD were differentiated based on the length of the infarction diameter and number of axial slices. The proximal/distal M1 diameter ratio was calculated. MCA geometrics between LD and BAD were compared. Independent factors associated with LD were investigated. Computational fluid dynamics (CFD) analysis was used to evaluate hemodynamic parameters.

### Results

A total of 117 patients were included, of whom 64 (54.7%) and 53 (45.3%) were classified as BAD and LD, respectively. LD was associated with hypertension and favorable prognosis. MCA geometric variables revealed that LD had a higher proximal/distal M1 diameter ratio, indicating a potential distinguishing factor. Multivariate analysis confirmed the independent association between LD and the proximal/distal M1 diameter ratio. The proximal/distal M1 diameter ratio also showed a positive correlation with the number of ipsilesional lacunes. CFD analysis showed that the

**Data availability statement:** All relevant data are within the manuscript and its Supporting Information files.

**Funding:** This research was supported by the Digital Therapeutics Development and Clinical Validation Program, funded by the Ministry of Science and ICT (MSIT) and the National IT Industry Promotion Agency (NIPA) of Korea (Grant Number: H0601-25-1046).

**Competing interests:** The authors have declared that no competing interests exist.

**Abbreviations:** ACA, anterior cerebral artery; BAD, branch atheromatous disease; DWI, diffusion-weighted imaging; FLAIR, fluid-attenuated inversion recovery; LD, lipohyalinotic degeneration; LSA, lenticulostriate artery; MCA, middle cerebral artery; NIHSS, National Institutes of Health Stroke Scale; SVD, small vessel disease.

LD model had faster, greater blood influx into LSAs and higher wall shear stress and pressure gradient compared with the BAD model.

## Conclusions

This study suggests MCA geometry, particularly the proximal/distal M1 diameter ratio, may serve as an independent factor for identifying LD.

## Introduction

Despite their identical anatomical locations, subcortical infarctions within the lenticulostriate artery (LSA) territory can exhibit various pathophysiological characteristics. They can be categorized into two main subtypes: lipohyalinotic degeneration (LD) and branch atheromatous disease (BAD). LD, or lipohyalinosis, presents as concentric thickening of the hyaline layer within cerebral small vessels, ultimately leading to vessel occlusion. It is caused by hypertension-induced vascular hypertrophy and fibrinoid degeneration of perforating arteries, resulting in small subcortical infarcts [1]. Conversely, BAD is related to occlusion or severe stenosis of the ostial segment of the perforating artery, primarily because of underlying pathological changes, including atherosclerotic plaque formation [2–4]. To the best of our knowledge, the causative factors associated with the occurrence of LD or BAD have not been clearly verified.

Geometric variations in cerebral vessels have been linked to the presence and severity of small vessel disease (SVD), encompassing conditions such as leukoaraiosis, lacunar stroke, and intracranial hemorrhage. Notably, increased vascular tortuosity is positively correlated with SVD burden [5–9]. The proposed mechanisms underlying how tortuosity influences SVD burden are related primarily to the hemodynamic stress generated by flow perturbation, which can lead to endothelial dysfunction and direct damage to perforators [10]. Several studies have highlighted the importance of the diameter ratio of the first segment of the middle cerebral artery (MCA) in increasing the risk of deep intracerebral hemorrhage and the development of cerebral aneurysms [11,12]. Specifically, an elevated diameter ratio of the first segment of the MCA (proximal/distal M1 diameter ratio) may affect wall pressure and blood velocity within perforators, potentially leading to damage within the LSA [11]. In the present study, we explored the associations between the two distinct pathomechanisms of LSA infarction and MCA geometry, with a particular focus on the proximal/distal M1 diameter ratio.

## Methods

### Patients

Patients with acute ischemic stroke who were admitted to Asan Medical Center within 7 days of symptom onset between January 2020 and December 2021 were included. The inclusion criteria were based on the classification of small vessel occlusion according to the Trial of ORG 10172 in Acute Stroke Treatment (TOAST) classification, along with confirmation of an acute ischemic lesion within the LSA

territory through diffusion-weighted imaging (DWI). Patients who demonstrated significant stenosis (>50% by diameter) or occlusion in the first segment of the MCA (M1) were excluded. Moreover, individuals who could not undergo magnetic resonance imaging (MRI) and magnetic resonance angiography for confirming acute ischemic stroke and measuring MCA geometrical variables and those deemed unsuitable for measurement because of poor imaging quality were excluded.

Demographic and clinical data, encompassing vascular risk factors, were retrospectively collected from a stroke registry using electronic medical record data. The results of serologic tests, conducted either on the day of admission or the day following admission after an 8-h fasting period, were used in the analysis. The National Institutes of Health Stroke Scale (NIHSS) score was measured to assess stroke severity upon admission and discharge, and the difference was calculated to assess neurological deterioration or improvement during admission. Neuroimaging was performed on the day of admission at the emergency medical center, following the center's established protocol (outlined below).

This study received approval from the Institutional Review Board (IRB) of Asan Medical Center, and informed consent requirements were waived because of the retrospective nature of the study (IRB number: 2023--1381). We have accessed the data from 2023 November to 2024 April.

### Neuroimaging

MRI scans were performed via either a 1.5-T Siemens Avanto scanner (Siemens Medical Solutions, Malvern, PA, USA) or a 3.0-T Philips scanner (Philips Healthcare, Eindhoven, The Netherlands). The imaging protocol comprised axial DWI, gradient echo or susceptibility-weighted images, fluid-attenuated inversion recovery (FLAIR) images, intracranial angiograms employing time-of-flight (TOF) images, and contrast-enhanced magnetic resonance angiograms, which encompassed the cervical arteries.

The geometric parameters of the MCA were measured from the M1 segment of the ipsilesional symptomatic MCA. The M1 segment was defined as the portion between the anterior cerebral artery (ACA) bifurcation point and the M2 bifurcation point [13]. The following parameters were assessed: proximal M1 diameter (the closest point measurable from the ACA–MCA bifurcation), distal M1 diameter (the closest point measurable from the M2 bifurcation point), actual length of M1 (centerline length of M1), and straight length of M1 (the shortest distance between the ACA–MCA bifurcation and the M2 bifurcation point). The proximal/distal M1 diameter ratio was calculated by dividing the proximal M1 diameter by the distal M1 diameter (proximal M1 diameter/distal M1 diameter × 100%, Fig 1). The tortuosity of M1 was calculated by

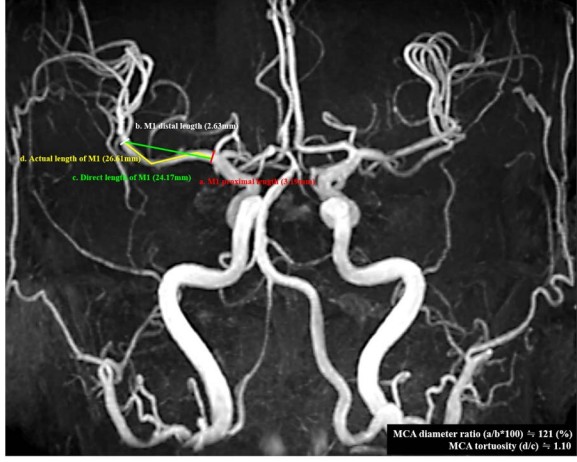

**Fig 1. Measurement of MCA variables.** The M1 diameter ratio is defined as a/b × 100 (%). M1 tortuosity is defined as d/c.

dividing the actual length of M1 by its straight length [14]. Although only patients with insignificant stenosis of the MCA (<50% by diameter) were included, the degree of stenosis was measured and classified as 0–10%, 10–30%, or 30–50%.

To ensure measurement reliability, all MCA geometric measurements were performed independently by two trained neurologists (J.S.Y. and J.Y.P.) who were blinded to the clinical information and stroke subtype classification. For each parameter (proximal M1 diameter, distal M1 diameter, actual and straight length of M1), measurements were obtained twice, and the average value was used for analysis. Inter-observer reliability was assessed using ICCs, which showed excellent agreement for all measurements (proximal M1 diameter: ICC = 0.92, 95% CI 0.88–0.95; distal M1 diameter: ICC = 0.90, 95% CI 0.86–0.94; M1 length: ICC = 0.89, 95% CI 0.84–0.93). In cases of substantial disagreement (>10% difference), a consensus was reached through joint review with a senior neuroradiologist (B.J.K.).

### Image analysis

We defined LD as follows: maximum diameter of infarction <15 mm and fewer than three consecutive axial slices of lesions with 5 mm thickness (or fewer than five consecutive axial slices with 3 mm thickness). In contrast, we defined BAD as follows: maximum diameter of infarction >15 mm or three or more consecutive axial slices of lesions with 5 mm thickness (or five or more consecutive axial slices with 3 mm thickness) [15,16]. A representative case of LD and BAD and measurements of the infarction diameter and axial slices are shown in Fig 2.

On the basis of FLAIR imaging, leukoaraiosis was graded from 0 to 3 according to the Fazekas scale of deep white matter changes. Microbleeds were defined as unambiguous homogenous round lesions with a diameter of ≤5 mm with signal loss as determined by gradient echo. The presence and total number of microbleeds were recorded [17]. All image analyses were performed by a neurologist and a neuroradiologist who were blinded to all the clinical data.

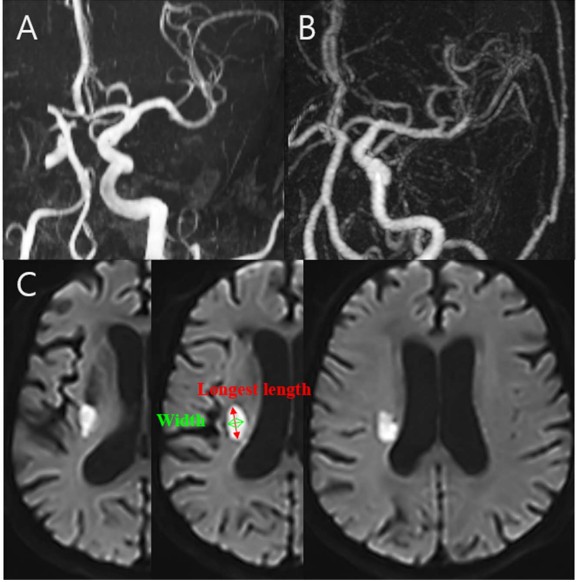

**Fig 2. Illustration of a representative case of lipohyalinotic degeneration (LD) and branch atheromatous disease (BAD).** (A) Angiographic findings of the left corona radiata LD showing a high M1 diameter ratio and tortuosity. (B) Angiographic findings of the left corona radiata BAD showing a low M1 diameter ratio and tortuosity. (C) Measurement of infarction size via maximal length and axial slices.

## Computational fluid dynamics (CFD) analysis

For the quantitative evaluation of hemodynamics, CFD analysis was conducted via ANSYS CFX (version 2020 R1). The fluid (blood) density and dynamic viscosity are 1050 kg/m$^3$ and 4.3 mPa·s, respectively. CFD simulations were conducted under steady laminar flow conditions. The parabolic velocity profile (Hagen–Poiseuille flow) was used as an inlet boundary condition for achieving the volumetric flow rate of the MCA (145 mL/min). On outlets, 0 Pa pressure was applied under the opening condition. The arterial walls were treated as rigid no-slip walls. A high-resolution advection scheme was used, and the convergence criterion of the numerical solutions was set at an RMS residual of $10^{-7}$. After the grid dependency study, tetrahedral grid systems including approximately 5.73 and 5.56 million mesh elements were selected for the LD and BAD models, respectively. We set up the LD and BAD models by using the actual numerical data from our results of the MCA diameter and length for the CFD calculation. Moreover, we used typical average values (angles of perforators = perpendicular to the MCA, numbers of perforators = 7, diameters of perforators = 0.5 mm) for data that could not be measured, including blood flow, blood viscosity, and the number and diameter of penetrating arteries. The LD model represents the MCA with a higher M1 diameter ratio, indicating a narrowing or conical MCA, whereas the BAD model represents a lower M1 diameter ratio, indicating a cylindrical MCA. We compared hemodynamic parameters, including blood velocity, blood flow rate, wall shear stress, and pressure gradient within the MCA and perforators.

## Statistical analysis

We conducted a comprehensive comparison of demographics; clinical data, including stroke risk factors; and radiological findings, including MCA geometric variables and SVD burden, between LD and BAD patients. To achieve this goal, we employed appropriate statistical tests, specifically Student's *t* tests for continuous variables and chi-square tests for categorical variables. Continuous variables are presented as the means along with their corresponding standard deviations, whereas categorical variables are expressed as percentages. The inclusion criterion for multivariate logistic regression analysis involved variables with p values less than 0.25 in the initial univariate analysis. To determine potential correlations among these variables, we conducted bivariate correlation analysis via Pearson's correlation coefficient, which is represented as rho ($\rho$). Regarding the proximal and distal diameter of MCA, intraclass correlation coefficient (ICC) was calculated. Receiver operating characteristic (ROC) curve analysis was performed to evaluate the discriminative ability of the M1 diameter ratio, with area under the curve (AUC), sensitivity, specificity, and an optimal cut-off value being reported. All the statistical analyses were performed via SPSS version 19 (IBM, Armonk, NY). Statistical significance was considered at $p < 0.05$.

## Results

We gathered data from 1625 patients who experienced acute ischemic stroke within 7 days of onset. In this cohort, 337 (20.7%) individuals were classified under the SVD category according to the TOAST classification criteria. After excluding patients with brainstem and thalamic SVDs, 117 (34.7%) patients with LSA infarction were ultimately included in the analysis. The mean age of the patients was 63.6 ± 11.8 years, and 75 (64.1%) patients were male. Within this group, 53 (45.3%) patients were categorized as having LD, whereas 64 (54.7%) were classified as having BAD.

### Comparison between BAD and LD

When comparing BAD and LD, we found that the two groups had a well-balanced age and sex distribution. LD was more strongly associated with hypertension than BAD was. Moreover, LD was associated with a lower incidence of dyslipidemia and a history of stroke. However, these differences did not reach statistical significance. In addition, LD patients had a more favorable prognosis, as evidenced by a lower discharge NIHSS score (2.3 ± 2.2 vs. 3.6 ± 2.6, p = 0.004) and a greater improvement in the NIHSS score during admission than BAD patients did (−0.7 ± 2.3 vs. −0.1 ± 1.8, p = 0.129).

In terms of MCA geometry-related variables, LD tended toward a smaller distal M1 diameter (2.1±0.3 vs. 2.2±0.4 mm, p=0.09) and a longer M1 length (23.2±9.9 vs. 20.9±7.7 mm, p=0.163) than BAD did, but these differences did not reach statistical significance. Moreover, the LD group had a significantly greater proximal/distal M1 diameter ratio than the BAD group did (120.9±20.7 vs. 110.4±16.9 (%), p=0.003). Conversely, the BAD tended to have a straighter M1 shape and less tortuosity than the LD. When we compared the degree of stenosis of the ipsilesional MCA, the two groups did not differ (Table 1). Furthermore, the proximal/distal M1 diameter ratio was strongly associated with the presence of ipsilateral lacunes (1.8±0.9 vs. 1.4±0.7, p=0.038).

### Factors associated with LD

In the univariate analysis, the discharge NIHSS score (odds ratio (OR) = 0.788, 95% confidence interval (CI) = 0.666–0.933, p=0.006), number of ipsilesional hemisphere lacunes (OR = 1.641, 95% CI = 1.203–2.631, p=0.040), and proximal/distal M1 diameter ratio (%) (OR = 1.031, 95% CI = 1.009–1.052, p=0.005) were associated with LD.

On the basis of the results of multivariate analysis, previous stroke history (OR = 0.046, 95% CI 0.003–0.642, p=0.022), discharge National Institutes of Health Stroke Scale (NIHSS) score (OR = 0.705, 95% CI 0.572–0.868, p=0.001), and the proximal/distal M1 diameter ratio (OR = 1.027, 95% CI 1.004–1.052, p=0.024) were independent factors associated with LD after adjusting for other confounding factors (Table 2).

### Proximal/distal M1 diameter ratio and SVD markers

The proximal/distal M1 diameter ratio was positively correlated with the number of lacunes in the ipsilesional hemisphere (Fig 3). Unfortunately, no associations were identified with other indicators, including the number of microbleeds, presence of intracerebral hemorrhage, or severity of leukoaraiosis. Furthermore, no significant correlation was observed with either initial stroke severity or stroke prognosis.

### ROC Analysis and Cut-off Value

To assess the diagnostic utility of the proximal/distal M1 diameter ratio in distinguishing LD from BAD, we performed receiver operating characteristic (ROC) curve analysis (Fig 4). The proximal/distal M1 diameter ratio demonstrated poor discriminative ability with an area under the curve (AUC) of 0.665 (95% CI 0.565–0.764, p=0.002). The optimal cut-off value we used was 112.5%, yielding both sensitivity and specificity of 62.3% for identifying LD.

### Hemodynamic comparison between the LD and BAD models

We used CFD analysis to compare the hemodynamic parameters. The LD model showed faster and greater influx of blood into the LSAs. Compared with the BAD perforators, the LD perforators had a 143% faster mean flow velocity (3.3 vs. 2.3, cm/s) and 144% higher mean flow rate (0.39 vs. 0.27, mL/min). The wall shear stress within the MCA and perforators was also greater in the LD model (Fig 5A and B), and a greater pressure gradient was calculated between the inlet and outlet of the MCA and LSAs (Fig 5C). In streamlines of blood flow, more eddying flow at the inlet of LSAs was observed in the BAD model than in the LD model (Fig 5A).

## Discussion

In the present study, we conducted a comprehensive comparison of the characteristics of two distinct types of ischemic strokes occurring within the LSA territory. Upon categorizing LSA infarctions into LD and BAD, we observed that the proximal/distal M1 diameter ratio was significantly greater in individuals with LD than in those with BAD. Importantly, our analysis revealed that a high proximal/distal M1 diameter ratio was an independent predictor of LD. Moreover, we observed that the proximal/distal M1 diameter ratio was significantly correlated with the number of lacunes, which is considered a radiological neuroimaging biomarker indicative of SVD.

**Table 1. Characteristics between patients with LD and BAD.**

| | Lipohyalinotic degeneration (n = 53) | Branch atheromatous disease (n = 64) | p |
|---|---|---|---|
| Demographic | | | |
| Age (years) | 64.0 ± 11.1 | 63.3 ± 12.5 | 0.733 |
| Gender, male (%) | 35 (66.0) | 40 (62.5) | 0.691 |
| Risk factor | | | |
| Hypertension (%) | 43 (81.1) | 43 (67.2) | 0.089 |
| Diabetes mellitus (%) | 15 (28.3) | 20 (31.3) | 0.729 |
| Dyslipidemia (%) | 24 (45.3) | 38 (59.4) | 0.128 |
| History of stroke (%) | 1 (1.9) | 6 (9.4) | 0.089 |
| History of smoking (%) | 28 (52.8) | 28 (43.8) | 0.328 |
| Body mass index (kg/m$^2$) | 24.0 ± 3.3 | 24.4 ± 2.7 | 0.483 |
| Clinical finding | | | |
| Initial NIHSS score | 3.0 ± 2.8 | 3.7 ± 2.9 | 0.205 |
| Discharge NIHSS score | 2.3 ± 2.2 | 3.6 ± 2.6 | 0.004 |
| NIHSS score change during admission | −0.7 ± 2.3 | −0.1 ± 1.8 | 0.129 |
| Discharge mRS | 1.5 ± 1.3 | 2.0 ± 1.1 | 0.037 |
| MCA geometric feature | | | |
| M1 proximal diameter (mm) | 2.5 ± 0.4 | 2.4 ± 0.4 | 0.340 |
| M1 distal diameter (mm) | 2.1 ± 0.3 | 2.2 ± 0.4 | 0.090 |
| M1 diameter ratio (%) | 120.9 ± 20.7 | 110.4 ± 16.9 | 0.003 |
| M1 shape | | | |
| Straight | 17 (32.1) | 27 (42.2) | 0.261 |
| Curved | 36 (67.9) | 39 (57.8) | |
| Inverted U-shape | 18 (34.0) | 17 (26.6) | |
| U-shape | 10 (18.9) | 9 (10.9) | |
| S-shape | 8 (15.1) | 13 (20.3) | |
| M1 centerline length (mm) | 23.2 ± 9.9 | 20.9 ± 7.7 | 0.163 |
| M1 tortuosity | 1.2 ± 0.2 | 1.1 ± 0.2 | 0.245 |
| M1 stenotic degree | | | 0.955 |
| 0–10% | 49 | 59 | |
| 11–30% | 2 | 3 | |
| 31–50% | 2 | 2 | |
| Small vessel burden (ipsilateral) | | | |
| Leukoaraiosis | | | 0.660 |
| Fazeka 0 | 3 (5.7) | 5 (7.8) | |
| Fazeka 1 | 24 (45.3) | 32 (50.0) | |
| Fazeka 2 | 16 (30.2) | 20 (31.3) | |
| Fazeka 3 | 10 (18.9) | 7 (10.9) | |
| No. of microbleeds | 0.3 ± 0.9 | 0.4 ± 1.1 | 0.543 |
| No. of lacunes | 1.8 ± 0.9 | 1.4 ± 0.7 | 0.038 |
| No. of ICH | 0.02 ± 0.14 | 0.02 ± 0.13 | 0.894 |

Data are presented as mean ± standard deviation, median (IQR), or number (%).

NIHSS, National Institutes of Health Stroke Scale; mRS, modified Rankin Scale; MCA, middle cerebral artery; ICH, intracerebral hemorrhage.

Pearson's chi-squared tests or Student's *t*-tests were used as appropriate.

**Table 2. Logistic regression analysis of factors associated with LD.**

| Variables | Unadjusted OR | Adjusted OR | p |
|---|---|---|---|
| HTN | 2.100 (0.885–4.980) | 1.663 (0.614–4.503) | 0.317 |
| Dyslipidemia | 0.566 (0.271–1.182) | 0.500 (0.207–1.205) | 0.122 |
| History of stroke | 0.186 (0.022–1.596) | 0.046 (0.003–0.642) | 0.022 |
| Discharge NIHSS | 0.788 (0.666–0.933) | 0.705 (0.572–0.868) | 0.001 |
| M1 diameter ratio (%) | 1.031 (1.009–1.052) | 1.027 (1.004–1.052) | 0.024 |
| M1 Tortuosity | 3.635 (0.402–32.850) | 1.345 (0.121–14.937) | 0.809 |
| Number of Lacunes | 1.641 (1.203–2.631) | 1.801 (0.982–3.304) | 0.057 |

Data are presented as odds ratios (95% confidence interval).

HTN, hypertension, NIHSS, National Institutes of Health Stroke Scale.

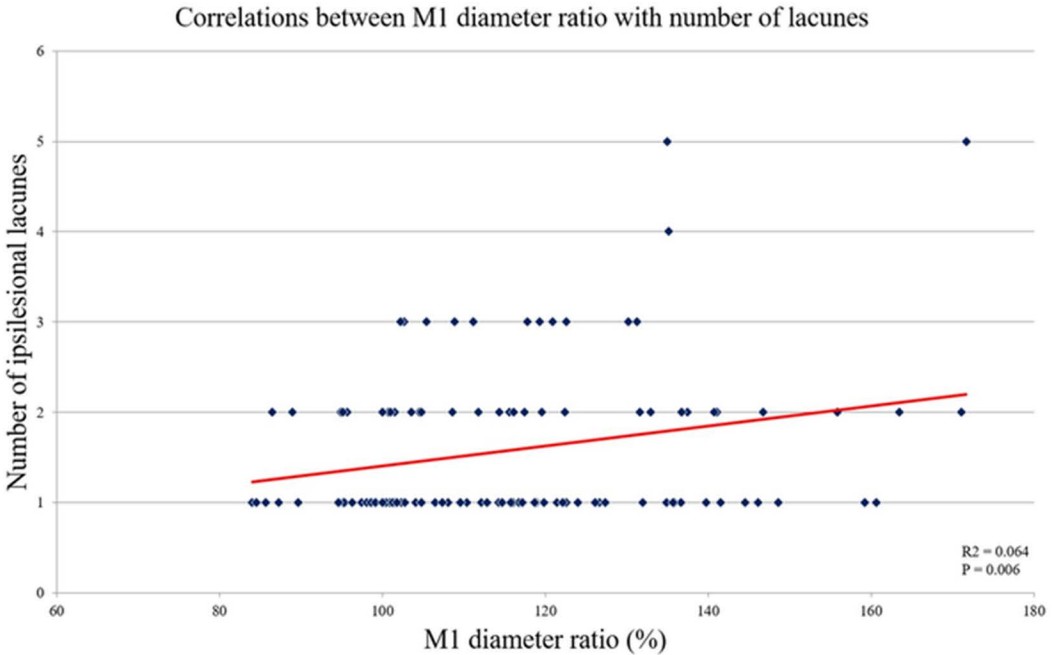

**Fig 3. Correlations between the M1 diameter ratio (%) and the number of lacunes.**

The correlation between the LD and elevated proximal/distal M1 diameter ratio can be explained by the principles of fluid dynamics. The cross-sectional area of the MCA is inversely related to the blood flow velocity according to the continuity equation. In simpler terms, a narrower distal MCA leads to accelerated blood flow, resulting in increased shear stress at the MCA and perforator walls (Fig 5B). Consequently, the increased shear stress could be attributed to heightened local hemodynamic damage to the perforators, ultimately leading to LD. Our CFD analysis revealed that, compared with BAD perforators, LD perforators presented faster velocities and greater shear stress, both of which could cause more damage within the perforators (Fig 5A). According to Bernoulli's principle, the pressure drop along the MCA is greater as the MCA proximal/distal diameter ratio is greater in the LD than in the BAD, which increases the flow and pressure at perforator inlets if it is assumed that the outlet pressure has identical or almost zero pressure conditions. A similar mechanism elucidates the link between the proximal/distal M1 diameter ratio and the presence of intracerebral hemorrhage [11].

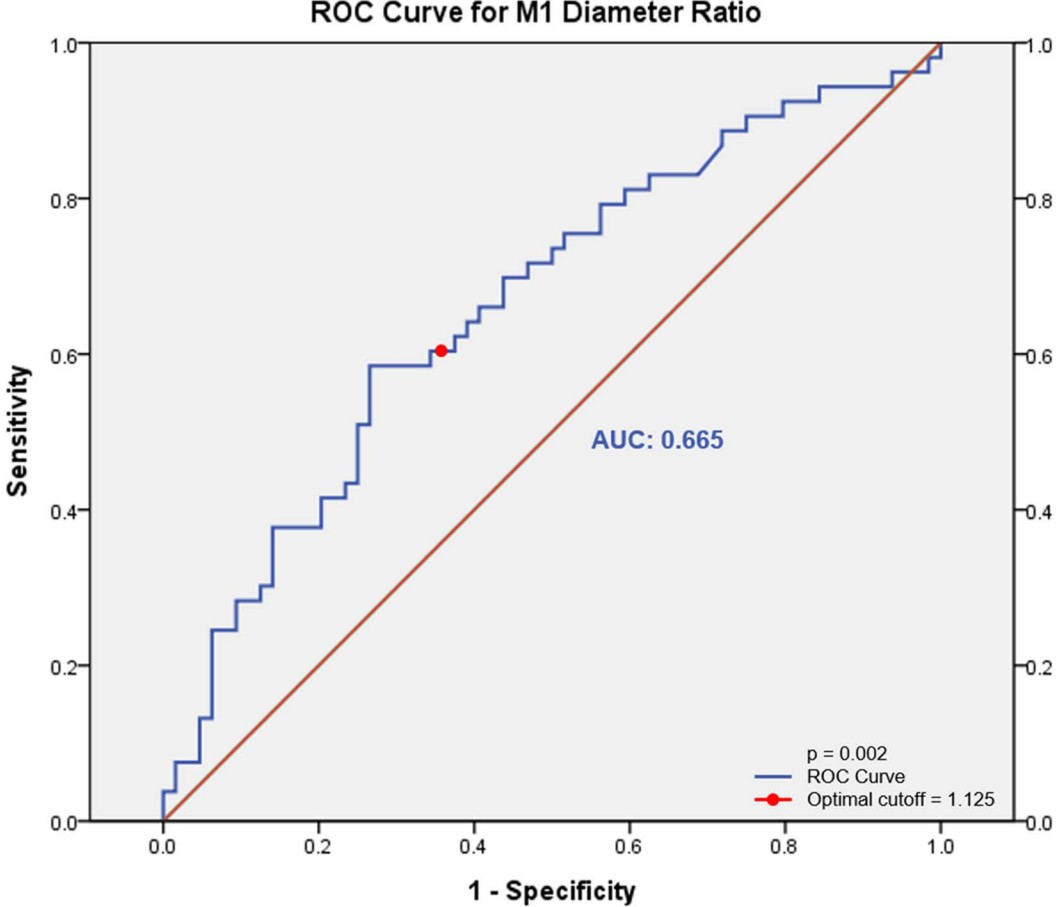

**Fig 4. Receiver operating characteristic (ROC) curve for the diagnostic performance of the M1 diameter ratio in distinguishing BAD from LD.**

The BAD model exhibited lower shear stress and eddying flows at the inlet of the perforators than did the LD model (Fig 5A and B). This finding is consistent with previous findings, as lower shear stress and eddying flows are known to be related to perforator orifice atherosclerosis progression [10,18,19] at the proximal inlets of perforators, which is the culprit of BAD.

Finally, while our ROC curve analysis shows poor discriminative ability with an AUC of 0.665, and moderate sensitivity and specificity at our cut-off value of 112.5% for the proximal/distal M1 diameter ratio, we can suggest it is not yet an adequate diagnostic method for distinguishing between LD and BAD alone (Fig 4). It should be considered as one component within a comprehensive diagnostic assessment rather than a standalone criterion. Future studies should investigate whether combining this geometric parameter with other clinical and radiological features could improve diagnostic accuracy.

Our study has several limitations. First, this was a retrospective observational study conducted at a single center. The single-center design and modest sample size (n = 117) of our study limit the generalizability of our findings. Cerebrovascular anatomy and risk factor profiles can vary significantly across different populations and ethnicities. Our findings should be considered preliminary until validated in larger, multicenter cohorts representing diverse populations. Such validation would be essential before considering the proximal/distal M1 diameter ratio as a clinically applicable biomarker for differentiating between LD and BAD mechanisms.

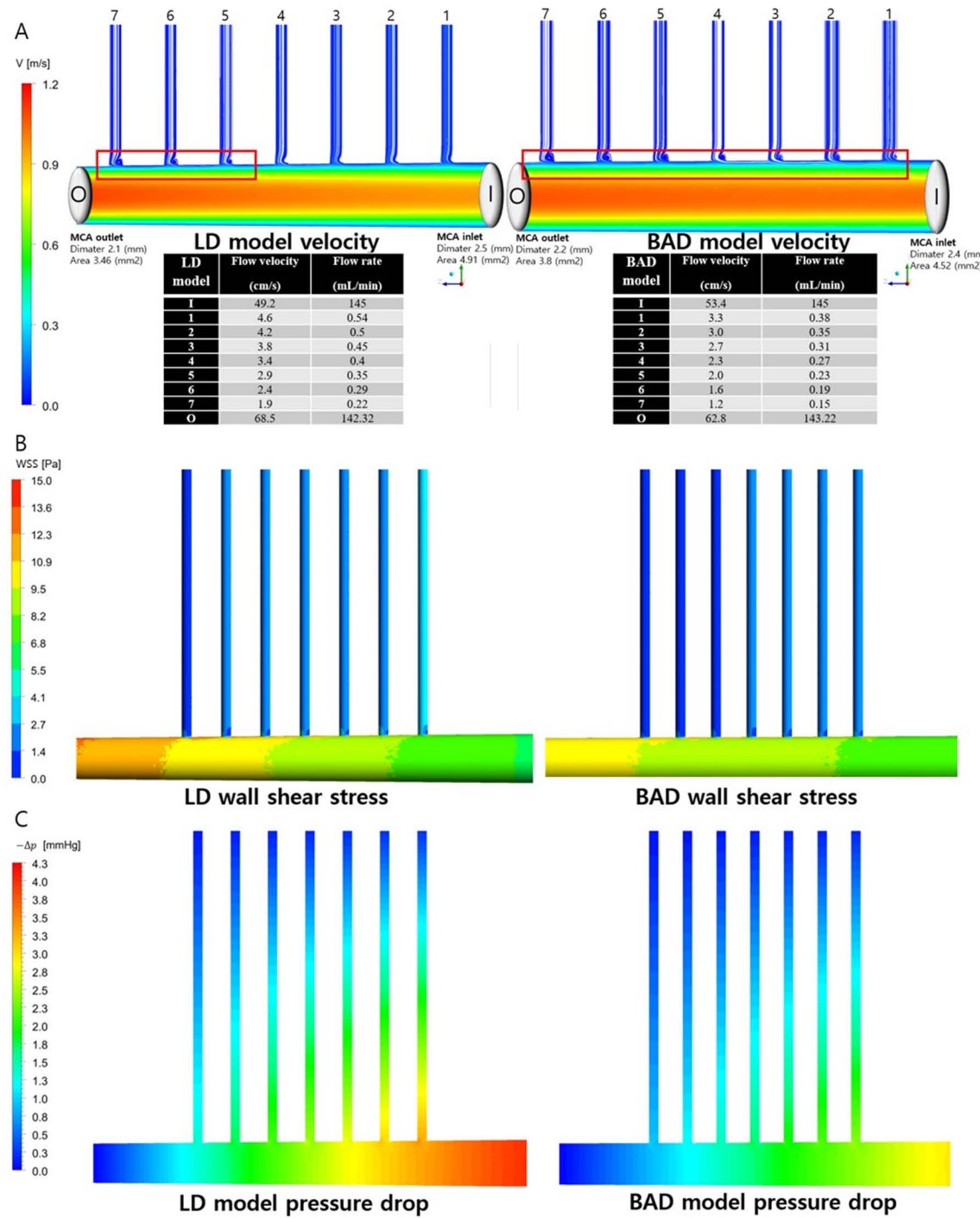

**Fig 5. Comparison of fluid hemodynamics within the MCA and lenticulostriate arteries in the LD/BAD model via computational fluid dynamics analysis.** (A) Velocities and flow rate. Eddying flows are marked in red rectangles. (B) Wall shear stress. (C) Pressure gradient.

Second major limitation in our study is the difficulty in definitively distinguishing between LD and BAD using conventional MRI TOF images. Advanced imaging techniques such as high-resolution vessel wall imaging could potentially provide more precise pathophysiological classification, by identifying culprit plaques or atheromas adjacent to perforator arteries. TOF-MRA has inherent limitations in detecting non-stenotic atherosclerotic plaques or those causing only mild luminal narrowing. Positive remodeling and early-stage atherosclerotic changes may be present without significant luminal narrowing, potentially leading to underestimation of atherosclerotic burden in both groups. In the absence of high-resolution modalities, we relied on established radiological criteria based on infarct size and axial slice involvement, which may have resulted in some degree of misclassification. However, TOF-MRA is currently the most widely used conventional tool in real-world clinical practice, and we believe that classification and analysis based on it will be of practical benefit. Future studies incorporating advanced vessel wall imaging techniques may help validate and refine our findings.

Finally, the assumptions for CFD, including the boundary conditions or unmeasurable values, may not appropriately reflect the actual conditions of cerebral blood flow, necessitating caution in interpretation. Our CFD simulations were based on idealized models using average measurements and simplified assumptions, such as steady-state flow and rigid arterial walls, which do not fully reflect the complex, dynamic nature of cerebral blood flow. These models did not incorporate individual anatomical variations, including the precise angle and number of perforators, variations in vessel diameter, or potential focal irregularities in the M1 segment. Furthermore, standardized parameters may overlook inter-individual differences in collateral flow patterns and cerebrovascular reactivity, both of which could significantly influence local hemodynamics. These limitations necessitate cautious interpretation of our findings and underscore the importance of future studies employing patient-specific CFD modeling with pulsatile flow, arterial wall elasticity, and high-resolution imaging to improve physiological accuracy and clinical relevance. In subsequent studies, we intend to employ 3D printing and particle image velocimetry [20].

Despite these limitations, our study demonstrated that an increased proximal/distal M1 diameter ratio, rather than BAD, is independently associated with LD. Furthermore, the number of lacunes is correlated with the M1 diameter ratio. This finding suggests that the proximal/distal M1 diameter ratio may impact hemodynamics, leading to LDs and lacunes.

## Supporting information

**S1 File. Data of this study.**
(XLSX)

## Author contributions

**Conceptualization:** Jun Sang Yoo, Jae Young Park, Jeong Yun Song, Jun Young Chang, Dong-Wha Kang, Sun U. Kwon, Hang Jin Jo.

**Data curation:** Jun Sang Yoo.

**Formal analysis:** Jun Sang Yoo, Bum Joon Kim, Jae Hyun Choi, Hang Jin Jo.

**Investigation:** Jun Sang Yoo, Jae Hyun Choi, Jae Young Park, Jeong Yun Song, Jun Young Chang, Hang Jin Jo.

**Methodology:** Bum Joon Kim, Hang Jin Jo.

**Supervision:** Bum Joon Kim, Jun Young Chang, Dong-Wha Kang, Sun U. Kwon, Hang Jin Jo.

**Validation:** Bum Joon Kim.

**Visualization:** Bum Joon Kim, Jae Hyun Choi, Hang Jin Jo.

**Writing – original draft:** Jun Sang Yoo.

**Writing – review & editing:** Jun Sang Yoo.

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
