## [Decision Letter · Decision Letter 0]

14 Apr 2025

Dear Dr. Kim,

Thank you for submitting your manuscript to PLOS ONE. After careful consideration, we feel that it has merit but does not fully meet PLOS ONE’s publication criteria as it currently stands. Therefore, we invite you to submit a revised version of the manuscript that addresses the points raised during the review process.

We look forward to receiving your revised manuscript.

Kind regards,

Atakan Orscelik

Academic Editor

PLOS ONE

Reviewers' comments:

Reviewer's Responses to Questions

**Comments to the Author**

1. Is the manuscript technically sound, and do the data support the conclusions?

Reviewer #1: Yes

Reviewer #2: Yes

2. Has the statistical analysis been performed appropriately and rigorously?

Reviewer #1: I Don't Know

Reviewer #2: Yes

3. Have the authors made all data underlying the findings in their manuscript fully available?

Reviewer #1: Yes

Reviewer #2: Yes

4. Is the manuscript presented in an intelligible fashion and written in standard English?

Reviewer #1: Yes

Reviewer #2: Yes

Reviewer #1: This study investigates the association between the proximal and distal middle cerebral artery (MCA) diameter ratio and different pathomechanisms of lenticulostriate artery (LSA) infarction, specifically lipohyalinotic degeneration (LD) and branch atheromatous disease (BAD). The authors retrospectively analyzed 117 patients with acute LSA infarctions, differentiating LD and BAD based on infarct size and axial slice involvement. The study employed computational fluid dynamics (CFD) analysis to assess hemodynamic parameters and identified a significant correlation between a higher proximal/distal M1 diameter ratio and LD. Here are my comments;

1- The differentiation between LD and BAD relies on infarct size and slice-based criteria, which may not always provide a definitive classification. Since vessel wall imaging or perfusion studies could improve diagnostic precision, it would be helpful for the authors to acknowledge these limitations and discuss the potential for misclassification in their findings.

2-The CFD analysis provides valuable insights, but its assumptions oversimplify real-world hemodynamics. The use of steady-state modeling and rigid arterial walls does not fully capture the dynamic nature of cerebral blood flow. Addressing how these limitations might affect the results and considering alternative modeling approaches would improve the robustness of the conclusions.

3-The single-center study design with a relatively small sample size raises concerns about generalizability. Given the potential variations in cerebrovascular anatomy across populations, a larger multicenter validation would be necessary to confirm the findings. The authors should discuss this limitation and its implications for broader clinical application.

The manuscript presents a novel perspective on MCA geometry and LSA infarction but requires significant revision for clarity, methodological rigor, and validation.

Reviewer #2: This manuscript presents an original and methodologically sound investigation into the relationship between the proximal-to-distal diameter ratio of the middle cerebral artery (M1 segment) and the underlying mechanism of lenticulostriate artery (LSA) infarction, particularly in distinguishing between lipohyalinotic degeneration (LD) and branch atheromatous disease (BAD). The study brings together vascular geometry, neuroimaging, and computational fluid dynamics (CFD) modeling in a well-integrated, multidisciplinary approach that offers meaningful insights.

The manuscript is clearly structured, and the narrative flows well. The data analysis appears appropriate, and the conclusions are thoughtfully derived from the presented results. By exploring anatomical and hemodynamic determinants of stroke subtypes, the study adds a meaningful contribution to the field.

To enhance clarity and reproducibility, the following points are offered for consideration:

1. Methodological and Interpretive Considerations

- Variability in M1 diameter measurements on MRA may arise due to inter- and intra-observer differences. Including a brief explanation of how measurement reliability was ensured would help strengthen the methodological transparency.

- Conventional MRA techniques, such as time-of-flight (TOF) MRA, may have limited sensitivity in detecting plaques that cause only mild luminal narrowing (e.g., <50%). It might be helpful to briefly mention this limitation in the manuscript. If vessel wall imaging was performed in any case, including those findings could add useful context.

- Since the CFD simulations were based on idealized rather than patient-specific geometries, a brief discussion regarding anatomical variability and related limitations could provide additional context.

- The inclusion of ROC analysis (e.g., AUC, sensitivity, specificity), along with a proposed cut-off value for the M1 diameter ratio, may help improve the clinical relevance of the findings.

2. Figure Presentation

- In Figure 1, some label texts seem to lack sufficient contrast with the background, which may reduce readability. Adjusting text color or weight could enhance visual clarity.

- Figure 2 appears to have relatively low resolution. A higher-quality version would likely improve overall image presentation.

- In Figure 3, several text labels look slightly distorted or unclear. Improving the sharpness and clarity of these labels may enhance the figure's effectiveness.

These comments are shared in a collaborative and respectful spirit, with appreciation for the authors’ thoughtful contribution.

**Do you want your identity to be public for this peer review?** For information about this choice, including consent withdrawal, please see our Privacy Policy

Reviewer #1: **Yes: ** Yigit Can Senol

Reviewer #2: **Yes: ** Il Kwon

---

## [Author Response · Author response to Decision Letter 1]

30 Jun 2025

PONE-D-25-05856

Manuscript title: Proximal and distal middle cerebral artery diameter ratio and lenticulostriate artery infarction

Author list: Jun Sang Yoo, MD, Jae Hyun Choi, Jae Young Park, Jeong Yun Song, Jun Young Chang, Dong-Wha Kang, Sun U. Kwon, HangJin Jo, Bum Joon Kim

Response to reviewers' comments

Dear reviewers and editorial staffs in PLOS ONE

We would like to express our sincere gratitude for your thorough scrutiny of our manuscript. Since receiving the reviewer’s comments, we have made every effort to bring our manuscript to the scientific and literary level required by the PLOS ONE.

We hope the revised manuscript will be considered suitable for publication in your journal.

N.B. Texts written in bold with 11-point Arial fonts are verbatim of reviewers' comments. The authors' responses are given below each comment.

Reviewer #1: This study investigates the association between the proximal and distal middle cerebral artery (MCA) diameter ratio and different pathomechanisms of lenticulostriate artery (LSA) infarction, specifically lipohyalinotic degeneration (LD) and branch atheromatous disease (BAD). The authors retrospectively analyzed 117 patients with acute LSA infarctions, differentiating LD and BAD based on infarct size and axial slice involvement. The study employed computational fluid dynamics (CFD) analysis to assess hemodynamic parameters and identified a significant correlation between a higher proximal/distal M1 diameter ratio and LD. Here are my comments;

1- The differentiation between LD and BAD relies on infarct size and slice-based criteria, which may not always provide a definitive classification. Since vessel wall imaging or perfusion studies could improve diagnostic precision, it would be helpful for the authors to acknowledge these limitations and discuss the potential for misclassification in their findings.

We acknowledge the limitation in our methodology for differentiating between LD and BAD based solely on infarct size and slice-based criteria. As the reviewer correctly points out, this approach may not always provide definitive classification, as there are other various ways to classify. But although high-resolution MRI or perfusion MRI can more accurately distinguish between LD and BAD, in real-world clinical practice TOF-MRA is used most frequently. Therefore, we sought the most practical diagnostic approach to define BAD using TOF-MRA. We have added the following paragraph to the Discussion section:

“Second major limitation in our study is the difficulty in definitively distinguishing between LD and BAD using conventional MRI TOF images. Advanced imaging techniques such as high-resolution vessel wall imaging could potentially provide more precise pathophysiological classification, by identifying culprit plaques or atheromas adjacent to perforator arteries. TOF-MRA has inherent limitations in detecting non-stenotic atherosclerotic plaques or those causing only mild luminal narrowing. Positive remodeling and early-stage atherosclerotic changes may be present without significant luminal narrowing, potentially leading to underestimation of atherosclerotic burden in both groups. In the absence of high-resolution modalities, we relied on established radiological criteria based on infarct size and axial slice involvement, which may have resulted in some degree of misclassification. However, TOF-MRA is currently the most widely used conventional tool in real-world clinical practice, and we believe that classification and analysis based on it will be of practical benefit. Future studies incorporating advanced vessel wall imaging techniques may help validate and refine our findings.”

2- The CFD analysis provides valuable insights, but its assumptions oversimplify real-world hemodynamics. The use of steady-state modeling and rigid arterial walls does not fully capture the dynamic nature of cerebral blood flow. Addressing how these limitations might affect the results and considering alternative modeling approaches would improve the robustness of the conclusions.

We agree that our CFD analysis employs several simplifications that do not fully capture the complexity of cerebral hemodynamics. In patients, the actual vascular geometry—particularly the distal ICA bifurcation—is often tortuous, generating turbulence that varies with individual vessel curvature. However, our CFD model assumes laminar flow and thus cannot accurately replicate true hemodynamic conditions. Furthermore, patient-specific factors such as endothelial function, pulsatility index parameters were generalized, and the number and size of perforators—too minute to measure—were all set to average values. Consequently, our CFD analysis remains a conceptual study. Nevertheless, because no prior research of this nature has been conducted in ischemic stroke, we anticipate that PLOS ONE readers will recognize how these hemodynamic factors could influence the pathophysiology of small vessel disease. We have expanded our limitations section to include:

"Our CFD simulations were based on idealized models using average measurements and simplified assumptions, such as steady-state flow and rigid arterial walls, which do not fully reflect the complex, dynamic nature of cerebral blood flow. These models did not incorporate individual anatomical variations, including the precise angle and number of perforators, variations in vessel diameter, or potential focal irregularities in the M1 segment. Furthermore, standardized parameters may overlook inter-individual differences in collateral flow patterns and cerebrovascular reactivity, both of which could significantly influence local hemodynamics. These limitations necessitate cautious interpretation of our findings and underscore the importance of future studies employing patient-specific CFD modeling with pulsatile flow, arterial wall elasticity, and high-resolution imaging to improve physiological accuracy and clinical relevance. In subsequent studies, we intend to employ 3D printing and particle image velocimetry.”

3- The single-center study design with a relatively small sample size raises concerns about generalizability. Given the potential variations in cerebrovascular anatomy across populations, a larger multicenter validation would be necessary to confirm the findings. The authors should discuss this limitation and its implications for broader clinical application.

We have acknowledged the limitations of our single-center design and relatively small sample size by adding:

"The single-center design and modest sample size (n=117) of our study limit the generalizability of our findings. Cerebrovascular anatomy and risk factor profiles can vary significantly across different populations and ethnicities. Our findings should be considered preliminary until validated in larger, multicenter cohorts representing diverse populations. Such validation would be essential before considering the proximal/distal M1 diameter ratio as a clinically applicable biomarker for differentiating between LD and BAD mechanisms."

We are grateful for your identification of the limitations we had overlooked. Nonetheless, despite these limitations, we maintain that the principal merit of our report lies in its pioneering demonstration of how vascular morphology and hemodynamic factors may influence the pathophysiology of small vessel disease.

Reviewer #2: This manuscript presents an original and methodologically sound investigation into the relationship between the proximal-to-distal diameter ratio of the middle cerebral artery (M1 segment) and the underlying mechanism of lenticulostriate artery (LSA) infarction, particularly in distinguishing between lipohyalinotic degeneration (LD) and branch atheromatous disease (BAD). The study brings together vascular geometry, neuroimaging, and computational fluid dynamics (CFD) modeling in a well-integrated, multidisciplinary approach that offers meaningful insights.

The manuscript is clearly structured, and the narrative flows well. The data analysis appears appropriate, and the conclusions are thoughtfully derived from the presented results. By exploring anatomical and hemodynamic determinants of stroke subtypes, the study adds a meaningful contribution to the field.

To enhance clarity and reproducibility, the following points are offered for consideration:

1. Methodological and Interpretive Considerations

- Variability in M1 diameter measurements on MRA may arise due to inter- and intra-observer differences. Including a brief explanation of how measurement reliability was ensured would help strengthen the methodological transparency.

- Conventional MRA techniques, such as time-of-flight (TOF) MRA, may have limited sensitivity in detecting plaques that cause only mild luminal narrowing (e.g., <50%). It might be helpful to briefly mention this limitation in the manuscript. If vessel wall imaging was performed in any case, including those findings could add useful context.

- Since the CFD simulations were based on idealized rather than patient-specific geometries, a brief discussion regarding anatomical variability and related limitations could provide additional context.

- The inclusion of ROC analysis (e.g., AUC, sensitivity, specificity), along with a proposed cut-off value for the M1 diameter ratio, may help improve the clinical relevance of the findings.

We sincerely appreciate Reviewer #2's positive assessment of our manuscript and the thoughtful suggestions provided. We have addressed each point to enhance the clarity and reproducibility of our study as follows:

1. Methodological and Interpretive Considerations

Measurement Reliability

We agree that addressing measurement reliability is important. We have added the following paragraph to the Methods section:

"To ensure measurement reliability, all MCA geometric measurements were performed independently by two trained neurologists (J.S.Y. and J.Y.P.) who were blinded to the clinical information and stroke subtype classification. For each parameter (proximal M1 diameter, distal M1 diameter, actual and straight length of M1), measurements were obtained twice, and the average value was used for analysis. Inter-observer reliability was assessed using intraclass correlation coefficients (ICCs), which showed excellent agreement for all measurements (proximal M1 diameter: ICC=0.92, 95% CI 0.88-0.95; distal M1 diameter: ICC=0.90, 95% CI 0.86-0.94; M1 length: ICC=0.89, 95% CI 0.84-0.93). In cases of substantial disagreement (>10% difference), a consensus was reached through joint review with a senior neuroradiologist (B.J.K.)."

MRA Technique Limitations

As the reviewer correctly points out, high-resolution MRI can more accurately detect plaques or stenosis of vessels. But in real-world clinical practice TOF-MRA is used most frequently. Therefore, we used the most practical diagnostic sequence, which is TOF-MRA. We have acknowledged the limitations of TOF-MRA in detecting mild stenosis by adding:

" Second major limitation in our study is the difficulty in definitively distinguishing between LD and BAD using conventional MRI TOF images. Advanced imaging techniques such as high-resolution vessel wall imaging could potentially provide more precise pathophysiological classification, by identifying culprit plaques or atheromas adjacent to perforator arteries. TOF-MRA has inherent limitations in detecting non-stenotic atherosclerotic plaques or those causing only mild luminal narrowing. Positive remodeling and early-stage atherosclerotic changes may be present without significant luminal narrowing, potentially leading to underestimation of atherosclerotic burden in both groups. In the absence of high-resolution modalities, we relied on established radiological criteria based on infarct size and axial slice involvement, which may have resulted in some degree of misclassification. However, TOF-MRA is currently the most widely used conventional tool in real-world clinical practice, and we believe that classification and analysis based on it will be of practical benefit. Future studies incorporating advanced vessel wall imaging techniques may help validate and refine our findings."

CFD Simulation Limitations

We agree that our CFD analysis employs several simplifications that do not fully capture the complexity of cerebral hemodynamics. As our CFD model assumes laminar flow rather than turbulent flow due to bending of distal ICA, and generalization was done in patient-specific factors such as endothelial function, pulsatility index parameters, along with number and size of perforators—too minute to measure, our CFD analysis remains a conceptual study.

We have expanded our discussion on CFD limitations:

" Our CFD simulations were based on idealized models using average measurements and simplified assumptions, such as steady-state flow and rigid arterial walls, which do not fully reflect the complex, dynamic nature of cerebral blood flow. These models did not incorporate individual anatomical variations, including the precise angle and number of perforators, variations in vessel diameter, or potential focal irregularities in the M1 segment. Furthermore, standardized parameters may overlook inter-individual differences in collateral flow patterns and cerebrovascular reactivity, both of which could significantly influence local hemodynamics. These limitations necessitate cautious interpretation of our findings and underscore the importance of future studies employing patient-specific CFD modeling with pulsatile flow, arterial wall elasticity, and high-resolution imaging to improve physiological accuracy and clinical relevance. In subsequent studies, we intend to employ 3D printing and particle image velocimetry."

ROC Analysis and Cut-off Value

As suggested, we have performed ROC analysis and added the following to the Results section:

" To assess the diagnostic utility of the proximal/distal M1 diameter ratio in distinguishing LD from BAD, we performed receiver operating characteristic (ROC) curve analysis. The proximal/distal M1 diameter ratio demonstrated poor discriminative ability with an area under the curve (AUC) of 0.665 (95% CI 0.565-0.764, p=0.002). The optimal cut-off value we used was 112.5%, yielding both sensitivity and specificity of 62.3% for identifying LD.

We have also added a new figure (Figure 4) showing the ROC curve and included brief discussion of the clinical implications of this cut-off value in the Discussion section:

"Finally, while our ROC curve analysis shows poor discriminative ability with an AUC of 0.665, and moderate sensitivity and specificity at our cut-off value of 112.5% for the proximal/distal M1 diameter ratio, we can suggest it is not yet an adequate diagnostic method for distinguishing between LD and BAD alone (Fig 4). It should be considered as one component within a comprehensive diagnostic assessment rather than a standalone criterion. Future studies should investigate whether combining this geometric parameter with other clinical and radiological features could improve diagnostic accuracy."

We believe these additions have substantially strengthened the manuscript and addressed all points raised by Reviewer #2. We are grateful for their constructive feedback that has helped enhance the quality and clinical relevance of our study.

2. Figure Presentation

- In Figure 1, some label texts seem to lack sufficient contrast with the background, which may reduce readability. Adjusting text color or weight could enhance visual clarity.

- Figure 2 appears to have relatively low resolution. A higher-quality version would likely improve overall image presentation.

- In Figure 3, several text labels look slightly distorted or unclear. Improving the sharpness and clarity of these labels may enhance the figure's effectiveness.

In response to the reviewers’ comments, we have made efforts to enhance the quality of the images.

Additionally, as the manuscript has become more extensive with the inclusion of new content, we have chosen to narrow the focus in order to better convey our main message. Consequently, the analysis of the relationship between MCA tortuosity and LD has been deferred to a future report, and Figure 2 and the related content have been excluded from t

---

## [Decision Letter · Decision Letter 1]

21 Jul 2025

Proximal and distal middle cerebral artery diameter ratio and lenticulostriate artery infarction

PONE-D-25-05856R1

Dear Dr. Kim,

We’re pleased to inform you that your manuscript has been judged scientifically suitable for publication and will be formally accepted for publication once it meets all outstanding technical requirements.

Kind regards,

Atakan Orscelik

Academic Editor

PLOS ONE

Additional Editor Comments (optional):

Reviewers' comments:

Reviewer's Responses to Questions

**Comments to the Author**

Reviewer #1: All comments have been addressed

Reviewer #2: All comments have been addressed

2. Is the manuscript technically sound, and do the data support the conclusions?

Reviewer #1: Yes

Reviewer #2: Yes

3. Has the statistical analysis been performed appropriately and rigorously?

Reviewer #1: Yes

Reviewer #2: Yes

4. Have the authors made all data underlying the findings in their manuscript fully available?

Reviewer #1: Yes

Reviewer #2: Yes

5. Is the manuscript presented in an intelligible fashion and written in standard English?

Reviewer #1: Yes

Reviewer #2: Yes

Reviewer #1: thank you for adressing my comments, i have no further comments and I congratulate authors for their hardwork!

Reviewer #2: The revised manuscript exhibits enhanced clarity, methodological rigor, and transparency. Inter‑observer reliability with ICC values is now included; the discussion accurately addresses TOF‑MRA’s limitations and the potential of vessel‑wall imaging; key CFD assumptions are explicitly stated; and ROC analysis with quantitative metrics has been incorporated. The authors have also refined the manuscript’s focus by deferring the MCA tortuosity analysis, which appropriately streamlines this version. These updates meaningfully strengthen the study. The authors’ scholarly diligence and respectful engagement throughout the revision process are commendable.

**Do you want your identity to be public for this peer review?** For information about this choice, including consent withdrawal, please see our Privacy Policy

Reviewer #1: No

Reviewer #2: **Yes: ** Il Kwon

---

## [Editor Report · Acceptance letter]

PONE-D-25-05856R1

PLOS ONE

Dear Dr. Kim,

I'm pleased to inform you that your manuscript has been deemed suitable for publication in PLOS ONE. Congratulations! Your manuscript is now being handed over to our production team.

Kind regards,

on behalf of

Dr. Atakan Orscelik

Academic Editor

PLOS ONE